# MONET: Debiasing Graph Embeddings via the Metadata-Orthogonal Training Unit

## Abstract

Are Graph Neural Networks (GNNs) fair? In many real world graphs, the formation of edges is related to certain node attributes (e.g. gender, community, reputation). In this case, standard GNNs using these edges will be biased by this information, as it is encoded in the structure of the adjacency matrix itself. In this paper, we show that when metadata is correlated with the formation of node neighborhoods, unsupervised node embedding dimensions learn this metadata. This bias implies an inability to control for important covariates in real-world applications, such as recommendation systems.

To solve these issues, we introduce the Metadata-Orthogonal Node Embedding Training (MONET) unit, a generalizable neural network architecture for performing training-time linear debiasing of graph embeddings. MONET achieves this by ensuring that the node embeddings are trained on a hyperplane orthogonal to that of the node metadata. This effectively organizes unstructured embedding dimensions into an interpretable topology-only, metadata-only division with no linear interactions. We illustrate the effectiveness of MONET though our experiments on a variety of real world graphs, which shows that our method can learn and remove the effect of arbitrary covariates in tasks such as preventing the leakage of political party affiliation in a blog network, and thwarting the gaming of embedding-based recommendation systems.

## 1 Introduction

Graph embeddings – continuous, low-dimensional vector representations of nodes – have been eminently useful in network visualization, node classification, link prediction, and many other graph learning tasks (10). While graph embeddings can be estimated directly by unsupervised algorithms using the graph's structure (e.g. 24; 28; 15; 25), there is often additional (non-relational) information available for each node in the graph. This information, frequently referred to as node *attributes* or node *metadata*, can contain information that is useful for prediction tasks including demographic, geo-spatial, and/or textual features.

The interplay between a node's metadata and edges is a rich and active area of research. Interestingly, in a number of cases, this metadata can be measurably related to a graph's structure (21), and in some instances there may be a causal relationship (the node's attributes influence the formation of edges). As such, metadata can enhance graph learning models (31; 20), and conversely, graphs can be used as regularizers in supervised and semi-supervised models of node features (32; 11). Furthermore, metadata are commonly used as evaluation data for graph embeddings (8). For example, node embeddings trained on a Flickr user graph were shown to predict user-specified Flickr "interests" (24). This is presumably because users (as nodes) in the Flickr graph tend to follow users with similar interests, which illustrates a potential causal connection between node topology and node metadata.

However, despite the usefulness and prevalence of metadata in graph learning, there are instances where it desirable to design a system to *avoid* the effects of a particular kind of *sensitive* data. For instance, the designers of a recommendation system may want to make recommendations independent of a user's demographic information or location.

At first glance, this may seem like an artificial dilemma – surely one could just avoid the problem by not adding such sensitive attributes to the model. However, such an approach (ignoring a sensitive

attribute) does not control for any existing correlations that may exist between the sensitive metadata and the edges of a node. In other words, if the edges of the graph are correlated with sensitive metadata, then any algorithm which does not explicitly model and remove this correlation will be biased as a result of it. Surprisingly, almost all of the existing work in the area (31; 35) has ignored this important realization.[1]

In this work, we seek to refocus the discussion about graph learning with node metadata. To this end, we propose a novel, general technique for extending graph representations with metadata embedding dimensions while debiasing the remaining (topology) dimensions. Specifically, our contributions are the following:

1. The Metadata-Orthogonal Node Embedding Training (MONET) unit, a novel GNN algorithm which jointly embeds graph topology and graph metadata while enforcing linear decorrelation between the two embedding spaces.

2. Analysis which proves that a naive approach (adding metadata embeddings without MONET) leaks metadata information into topology embeddings, and that the MONET unit does not.

3. Experimental results on real world graphs which show that MONET can successfully "debias" topology embeddings while relegating metadata information to separate metadata embeddings.

## 2 PRELIMINARIES

Early graph embedding methods involved dimensionality reduction techniques like multidimensional scaling and singular value decomposition (8). In this paper we use graph neural networks trained on random walks, similarly to DeepWalk (24). DeepWalk and many subsequent methods first generate a sequence of random walks from the graph, to create a "corpus" of node "sentences" which are then modeled via word embedding techniques (e.g. word2vec (19) or GloVe (23)) to learn low dimensional representations that preserve the observed co-occurrence similarity.

Let $W$ be a $d$-dimensional graph *embedding* matrix, $W \in \mathbb{R}^{n \times d}$, which aims to preserve the low-dimensional structure of a graph ($d << n$). Rows of $W$ correspond to nodes, and node pairs $i, j$ with large dot-products $W_i^T W_j$ should be structurally or topologically close in the graph. As a concrete example, in this paper we consider the debiasing of a recently proposed graph embedding using the GloVe model (6). Its training objective is:

$$\text{GloVe}(U, V, a, b | C) = \sum_{i,j \leq n} f_\alpha(C_{ij})(a_i + b_j + U_i^T V_j - \log(C_{ij}))^2, \qquad (1)$$

where $U, V \in \mathbb{R}^{n \times d}$ are the "center" and "context" embeddings, $a, b \in \mathbb{R}^{n \times 1}$ are the biases, $C$ is the walk-distance-weighted context co-occurrences, and $f_\alpha$ is the loss smoothing function (23). We use the GloVe model in the next section to illustrate topology/metadata embeddings and metadata-orthogonal training. However, the MONET unit we propose is broadly generalizable. To illustrate this, we also describe a MONET unit for DeepWalk (24), a popular graph embedding algorithm.

**Notation.** In this paper, given a matrix $A \in \mathbb{R}^{n \times d}$ and an index $i \in 1, \ldots, n$, $A_i$ denotes the $d \times 1$ $i$-th row vector of $A$. Column indices will not be used. $\mathbf{0}_{n \times d}$ denotes the $n \times d$ zero matrix, and $|| \cdot ||_F$ denotes the Frobenius norm.

## 3 METADATA EMBEDDINGS AND ORTHOGONAL TRAINING

In this section we present MONET, our proposed method for separating and controlling the effects of metadata on topology embeddings. First, we begin by outlining the straightforward extension of metadata to traditional embedding models in Section 3.1. Next, in Section 3.2, we prove that such a simple model will leak information from the metadata to the topology (structural) embeddings. Then, in Section 3.3 we present MONET, our proposed approach for training embeddings of a graph's

---

[1]While preparing this manuscript, we have become aware of a recent independent result (5) in this area for recommender graphs. In contrast to that work, we use a substantially different methodology which offers guarantees about the debiasing process.

structure which are not correlated with metadata. Finally, we conclude with some analysis of MONET in Section 3.4

## 3.1 JOINTLY MODELING METADATA & TOPOLOGY

A natural first approach to modeling the effects of metadata on the graph is to explicitly include the node metadata as part of a node embedding model. For instance, to extend Eq. (1), in addition to $U$ and $V$ (the "topology embeddings"), we can consider the node metadata $M$ directly ($M \in \mathbb{R}^{n \times m}$, row vector $M_i$ is the metadata for node $u_i$). We then can define metadata embeddings $X = MT_1$, $Y = MT_2$, where $T_1, T_2$ are trainable transformations, and propose the concatenations $[U, X]$ and $[V, Y]$ as full-graph representations. The GloVe loss with metadata embeddings is:

$$\text{GloVe}_{meta}(U, V, T_1, T_2, a, b | C, M) = \frac{1}{2} \sum_{i,j \leq n} f_\alpha(C_{ij})(a_i + b_j + U_i^T V_j + X_i^T Y_j - \log(C_{ij}))^2. \quad (2)$$

While in this paper we demonstrate metadata embeddings within the GloVe model, they can be incorporated in any dot-product-based graph neural network. For instance, the well-known DeepWalk (24) loss, which is based on word2vec (19), would incorporate metadata embeddings as follows:

$$\text{DeepWalk}_{meta}(U, V, T_1, T_2 | \mathcal{W}, M) = - \sum_{i,j \in \mathcal{W}} \log(U_i^T V_j + X_i^T Y_j) - \sum_{k \in K_i} \log(-U_i^T V_k - X_i^T Y_k).$$
$$(3)$$

Above, $\mathcal{W}$ is the set of context pairs from random walks, and $K_i$ is a set of negative samples associated with node $i$. For GloVe, DeepWalk, and many other GNNs, this approach augments the overall graph representation by concatenating metadata-learned dimensions.

However, this naïve approach does not guarantee that the topology embeddings converge to be decorrelated from the metadata embeddings. Suppose that the metadata (like demographic information) are indeed associated with the formation of links in the graph. In this case, any algorithm which does not explicitly model and remove the association will be biased as a result of it. In the next section we formalize this concept, which we call *metadata leakage*.

## 3.2 METADATA LEAKAGE IN GRAPH NEURAL NETWORKS

Here, we formally define *metadata leakage* for general topology and metadata embeddings, and show how it can occur even in embedding models with separate metadata embeddings. All proofs appear in the Appendix.

**Definition 1.** *The **metadata leakage** of metadata embeddings $Z \in \mathbb{R}^{n \times d_Z}$ into topology embeddings $W \in \mathbb{R}^{n \times d}$ is defined $\mathcal{ML}(Z, W) := ||Z^T W||_F^2$. We say that there is no metadata leakage if and only if $\mathcal{ML}(Z, W) = 0$.*

Without a more nuanced approach, metadata leakage can occur even in embedding models that explicitly include the metadata, like Eqs. (2) and (3). To demonstrate this, we consider for simplicity a reduced metadata-aware GloVe loss with $W := U = V \in \mathbb{R}^{n \times d}$ as the sole topology embedding and $T := T_1 = T_2 \in \mathbb{R}^{m \times d_Z}$ as the sole metadata transformation parameter. With $Z := MT$, the reduced loss is:

$$\text{GloVe}^*_{meta}(W, T, a | C, M) = \frac{1}{2} \sum_{i,j \leq n} (a_i + a_j + W_i^T W_j + Z_i^T Z_j - \log(C_{ij}))^2 \quad (4)$$

We now show that under a random update of the $\text{GloVe}^*_{meta}$ model in Eq. (4), the expected metadata leakage is non-zero. Specifically, let $(i, j)$ be a node pair from $C$, and define $\delta_W(i, j)$ as the incurred Stochastic Gradient Descent update $W' \leftarrow W + \delta_W(i, j)$. Suppose there is a "ground-truth" metadata transformation $B \in \mathbb{R}^{m \times d_B}$, and define ground-truth metadata embeddings $\tilde{Z} := MB$, which represent the "true" dimensions of the metadata effect on the co-occurrences $C$. Define $\Sigma_B := BB^T$ and $\Sigma_T := TT^T$. With expectations taken with respect to the sampling of a pair $(i, j)$ for Stochastic Gradient Descent, define $\mu_W := \mathbb{E}[W_i]$ and $\Sigma_W := \mathbb{E}[W_i W_i^T]$. Define $\mu_M, \Sigma_M$ similarly. Then our main Theorem is as follows:

---

**Algorithm 1** MONET Unit Training Step

---

Given: topology embedding $W$, metadata embedding $Z$
 1: **procedure** FORWARD PASS DEBIASING($W$, $Z$)
 2:     Compute $Z$ left-singular vectors $Q_Z$ and projection $P_Z \leftarrow I_{n \times n} - Q_Z Q_Z^T$
 3:     Compute orthogonal topology embedding $W^\perp \leftarrow P_Z W$
 4:     **return** debiased graph representation $[W^\perp, Z]$
 5: **procedure** BACKWARD PASS DEBIASING($\delta_W$)
 6:     Compute orthogonal topology embedding update $\delta_W^\perp \leftarrow P_Z \delta_W$
 7:     Apply update $W^\perp \leftarrow W^\perp + \delta_W^\perp$
 8:     **return** debiased topology embedding $W^\perp$

---

**Theorem 1.** *Assume $\Sigma_W = \sigma_W I_d$ for $\sigma_W > 0$, $\mu_W = \mathbf{0}_{d \times 1}$, and $\mu_M = \mathbf{0}_{m \times 1}$. Suppose for some fixed $\theta \in \mathbb{R}$ we have $\log(C_{ij}) = \theta + \tilde{Z}_i^T \tilde{Z}_j$. Let $(i, j)$ be a randomly sampled co-occurrence pair and $W'$ the incurred update. Then if $\mathbb{E}[M_i W_i^T] = \beta \in \mathbb{R}^{m \times d}$, we have*

$$\mathbb{E}[\mathcal{ML}(Z, W')] \geq 2||T^T \left[\Sigma_M(\Sigma_B - \Sigma_T) + (n - \sigma_W)I_m\right]\beta||_F^2. \tag{5}$$

Importantly, $\Sigma_T$ and $\sigma_W$ are neural network hyperparameters, so we give a useful Corollary:

**Corollary 1.** *Under the assumptions of Theorem 1, $\mathbb{E}[\mathcal{ML}(Z, W)] = \Omega(n||T^T \beta||_F^2)$ as $n \to \infty$.*

Note that under reasonable GNN initialization schemes, $T$ and $\beta$ are random perturbations. Thus, Corollary 1 implies the surprising result that incorporating feed-forward metadata embeddings is not sufficient to prevent metadata leakage in practical settings.

### 3.3   MONET: METADATA-ORTHOGONAL NODE EMBEDDING TRAINING

Here, we introduce the Metadata-Orthogonal Node Embedding Training (MONET) unit for training joint topology-metadata graph representations $[W, Z]$ without metadata leakage. MONET explicitly prevents the correlation between topology and metadata, by using the Singular Value Decomposition (SVD) of $Z$ to *orthogonalize* updates to $W$ during training.

**MONET.** The MONET unit is a two-step algorithm applied to the training of a topology embedding in a neural network, and is detailed in Algorithm 1. The input to a MONET unit is a metadata embedding $Z \in \mathbb{R}^{n \times d_z}$ and a target topology embedding $W \in \mathbb{R}^{n \times d}$ for debiasing. Then, let $Q_Z$ be the left-singular vectors of $Z$, and define the projection $P_Z := I_{n \times n} - Q_Z Q_Z^T$. In the forward pass procedure, debiased topology weights are obtained by using the projection $W^\perp = P_Z W$. Similarly, $W^\perp$ is used in place of $W$ in subsequent GNN layers. In the backward pass, MONET also debiases the backpropagation update to the topology embedding, $\delta_W$, using $\delta_W^\perp = P_Z \delta_W$. Figure 1 illustrates a geometric interpretation of the MONET algorithm.

Straightforward properties of the SVD show that MONET directly prevents metadata leakage:

**Theorem 2.** *Using Algorithm 1, $\mathcal{ML}(Z, W^\perp) = 0$ and $\mathcal{ML}(Z, \delta^\perp) = 0$.*

We note that in this work we have only considered *linear* metadata leakage; debiasing nonlinear topology/metadata associations is an area of future work.

**Implementation (MONET$_G$ and MONET$_D$).** We demonstrate MONET in our experiments by applying Algorithm 1 to Eq. (2) and Eq. (3). We denote these models respectively by MONET$_G$ and MONET$_D$, for MONET "GloVe" and "DeepWalk". For MONET$_G$, we orthogonalize the input and output topology embeddings $U, V$ with the summed metadata embeddings $Z := X + Y$. By linearity, this implies $Z$-orthogonal training of the summed topology representation $W = U + V$. We note that working with the sums of center and context embeddings is the standard way to combine these matrices (23). Figure 2 shows an illustration of MONET$_G$. MONET$_D$ is implemented similarly, and is fully described in the Appendix Section A.3.

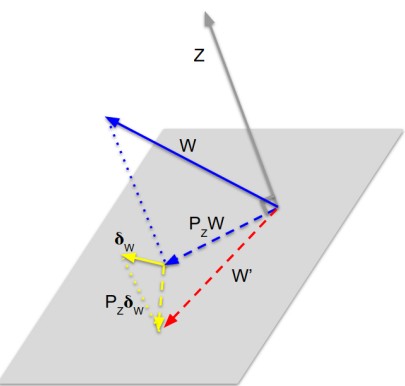

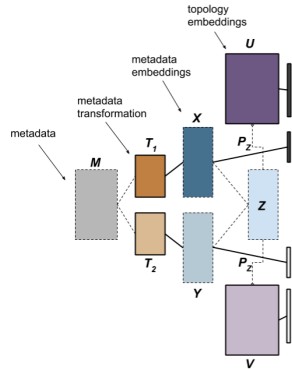

Figure 1: Geometric interpretation of MONET. Both prediction and training for $W$ occur on a hyperplane orthogonal to $Z$. In the forward pass, $W$ is projected onto the $Z$-orthogonal plane. When an update $\delta_W$ is proposed, it too is projected, resulting in the best metadata-orthogonal update. This allows $W$ to explore the space of unknown latent structure without bias from $Z$.

Figure 2: Illustration of MONET$_G$. $U$ and $V$ are topology embeddings. The MONET unit adds a feed-forward transformation of the metadata, resulting in metadata embeddings $X$ and $Y$. $Z = X + Y$ gives the combined metadata representation, used to debias $U$ and $V$ via $P_Z$. Dotted lines indicate stopped gradient flow during backpropagation.

### 3.4 ANALYSIS

Here we address some brief remarks about the algorithmic complexity of MONET, and the interpretation of its parameters.

**Algorithmic Complexity.** The bottleneck of MONET occurs in the SVD computation and orthogonalization. In our setting, the SVD is $O(nd_z^2)$ (29). The matrix $P_Z$ need not be computed to perform orthogonalization steps, as $P_Z W = W - Q_Z(Q_Z^T W)$, and the right-hand quantity is $O(ndd_z)$ to compute. Hence the general complexity of the MONET unit is $O(nd_z \max\{d, d_z\})$. In Table 3 we compare the wall clock time of MONET and baselines, showing only about a 10% overall time increase from standard GloVe.

**Metadata Parameter Interpretation.** The $i, j$ terms in the sum of the loss for GloVe models with metadata (GloVe$_{meta}$ and MONET$_G$) involve the dot product $X_i^T Y_j = M_i^T T_1 T_2^T M_j$. That expansion suggests that the matrix $\Sigma_T := T_1 T_2^T$ contains all pairwise metadata dimension relationships. In other words, $\Sigma_T$ gives the direction and magnitude of the raw metadata effect on log co-occurrence, and is therefore a way to measure the extent to which the model has captured metadata information. We will refer to this interpretation in the experiments that follow. An important experiment will show that applying the MONET algorithm increases the magnitude of $\Sigma_T$ entries.

**Connection to Adversarial Methods.** There are now many methods for supervised learning that use adversarial networks to produce data representations that are invariant to given factors (e.g. 30; 13). Because we are introducing MONET in the unsupervised graph learning setting, none of these approaches (to our knowledge) apply out-of-the-box to produce a baseline. To give an idea for how an adversarial approach might work as an alternative to MONET, we craft a adversarial version of GloVe which attempts to predict the metadata from the topology embeddings. This method, which to our knowledge is novel, is fully described in Section 4.1 and the Appendix.

## 4 METADATA DEBIASING EXPERIMENTS

Here we empirically demonstrate Theorems 1 and 2 by confirming the following hypotheses:

1. H1. The MONET unit can remove leakage of metadata information from topology embeddings, so that the topology embeddings cannot predict the metadata.

2. H2. The MONET unit can make recommender systems more robust to abuse by removing malicious user directions from rating graphs.

For all embedding models, we use the center-context embedding sum of topology embeddings $W := U + V$ as the graph representation for task evaluation. Note that some standard baselines (e.g. DeepWalk) do not incorporate metadata and therefore only train topology embeddings. All GloVe-based models are trained with TensorFlow (1) using the AdaGrad optimizer (12) with initial learning rate 0.05. DeepWalk models were trained using the gensim software (26).

### 4.1 QUANTITATIVE EXPERIMENT: POLITICAL BLOGS NETWORK

To address H1, illustrating Theorem 1 and the effect of MONET debiasing, we embed the effect of political ideology on a blogger network (3). The political blog network[2] has has 1,107 nodes corresponding to blog websites, 19,034 hyperlink edges between the blogs (after converting the graph to be undirected), and two clearly defined, equally sized communities of liberal and conservative bloggers.

**Methods and Design**. In this experiment all graph neural network models were trained on 5 iterations across 80 random walks per node of length 40 with context window size 10 ($MONET_D$ was trained on 20 iterations and used 5 negative samples per positive). Topology embeddings had dimension 16, and metadata embeddings had dimension 2. As one baseline, we use a random embedding generated from a 16-dimensional multivariate Normal. As our adversarial baseline, we apply the ideas introduced in (14) by adding a 2-layer MLP adversary to the GloVe model, referred to as $GloVe_{adversary}$. The adversary is trained to predict political party from the topology embeddings (more detail given in Appendix A.4).

We measure embedding bias by the Macro-F1 score of a linear SVM predicting political party from the embeddings, using a LIBLINEAR implementation (7). For each embedding set, we compute the mean and standard deviation Macro-F1 over 10 independent classification repetitions, each trained using half of the node labels sampled at random. To assess metadata information leakage, we also track the metadata dimension importance matrix $\Sigma_T := T_1 T_2^T$, recalling its interpretation from Section 3.

**Results**. Table 1 shows that the baselines DeepWalk and GloVe are highly effective at predicting political party, and therefore biased. This is unsurprising, as these methods are trained without metadata information, and were originally intended to encode low-dimensional structure like that present in this data set. The bias in DeepWalk and GloVe embeddings is further seen in their metadata leakage values, computed using political party one-hot vectors as metadata embeddings.

Considering the embedding models with metadata embeddings, we find that, interestingly, $GloVe_{meta}$'s topology embeddings are still able to predict political party with 88.3% Macro-F1. Also, as predicted by Corollary 1, $GloVe_{meta}$'s metadata leakage remains $O(n)$. This shows that simply concatenating metadata embeddings is not sufficient to isolate the metadata effect. In contrast, $MONET_G$ and $MONET_D$ achieve random Macro-F1 and no metadata leakage (under machine precision), demonstrating that on this data, the MONET unit is necessary to debias the blog embeddings from political party. Surprisingly, the MONET-enhanced models show significantly less bias than random embeddings, reflecting the fact that even random embeddings will not be perfectly linearly de-correlated from any given sensitive attribute.

The contrast between $GloVe_{meta}$ and $MONET_G$/$MONET_D$ is seen in two other ways. First, there is a noticeable increase in $\Sigma_T$ magnitude when MONET is used, implying that $GloVe_{meta}$ metadata embeddings are not capturing all possible metadata information. Second, as seen in Fig. 3, the 2-dimensional PCA plots of the $GloVe_{meta}$ embeddings still show political party separation, whereas the $MONET_G$ PCA dimensions reveal strong mixing.

In addition to the metrics in Table 1, we show additional metrics from this experiment in Appendix Table 3. In particular, that table contains results from a *non-linear* SVM applied to all embeddings. Accuracy from that classifier is high even on MONET embeddings, which emphasizes the fact that MONET only performs linear debiasing. We also report wall times for each method, and the embdding pairwise distance correlation to the GloVe model. These metrics (respectively) show that the SVD correction does not add substantial runtime, and that it does not overly corrupt the GloVe embedding signal.

---

[2]Available within the Graph-Tool software (22)

| Model | F1 (mean $\pm$ std) | $\Sigma_T = T_1 T_2^T$ (mean $\pm$ std) | $\mathcal{ML}$ |
|---|---|---|---|
| Random | 53.23% $\pm$0.73% | N/A | N/A |
| DeepWalk | 95.59% $\pm$0.07% | N/A | $2743.9 \pm 36.7$ |
| GloVe | 95.94% $\pm$0.07% | N/A | $6598.0 \pm 200.1$ |
| GloVe$_{adversary}$ | 81.46% $\pm$4.96% | N/A | $4459.0 \pm 430.4$ |
| GloVe$_{meta}$ | 88.33% $\pm$0.60% | $\begin{pmatrix} 0.108 \pm 0.006 & -0.106 \pm 0.004 \\ -0.108 \pm 0.009 & 0.106 \pm 0.006 \end{pmatrix}$ | $1827.6 \pm 289.7$ |
| MONET$_D$ | **48.60%** $\pm$0.50% | $\begin{pmatrix} 0.144 \pm 0.005 & -0.140 \pm 0.007 \\ -0.145 \pm 0.06 & 0.140 \pm 0.006 \end{pmatrix}$ | $0.001 \pm 0.001$ |
| MONET$_G$ | **49.30%** $\pm$0.60% | $\begin{pmatrix} 0.180 \pm 0.006 & -0.178 \pm 0.006 \\ -0.181 \pm 0.008 & 0.179 \pm 0.006 \end{pmatrix}$ | $0.018 \pm 0.002$ |

Table 1: Macro-F1 scores from political blog network classifications using graph topology embeddings only. MONET is successful in removing all metadata information from the topology embeddings – the links in the graph are no longer an effective predictor of political party. Comparison of the metadata transformation product $\Sigma_T$ between GloVe$_{meta}$ and MONET$_G$ shows MONET allows for considerably more metadata information learning. Finally, only MONET removes metadata leakage to precision error (recall $\mathcal{ML}()$ is a Frobenius norm).

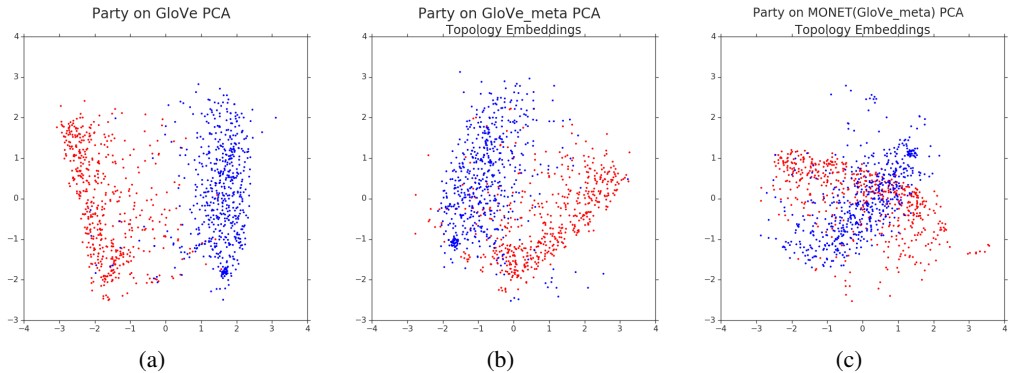

(a)      (b)      (c)

Figure 3: PCA of political blog graph embeddings. (a): Party separation clearly visible on standard GloVe embeddings. (b): Party separation reduces when GloVe$_{meta}$ captures some metadata information. (c): Party separation disappears with MONET$_G$ orthogonalized training.

| Model | Manipulated Items in Top-20 (mean $\pm$ std dev) | Embedding Distance Correlation w/GloVe |
|---|---|---|
| DeepWalk | $9.9 \pm 0.40$ | $0.385 \pm 0.005$ |
| GloVe | $9.8 \pm 0.76$ | $1.000 \pm 0.000$ |
| GloVe$_{meta}$ | $9.8 \pm 0.476$ | $0.852 \pm 0.002$ |
| NLP Debiasing (27; 4) (sum) | $8.1 \pm 1.7$ | $0.566 \pm 0.004$ |
| NLP Debiasing (27; 4) (max) | $4.9 \pm 3.4$ | **0.990** $\pm 0.003$ |
| MONET$_D$ | $5.7 \pm 2.0$ | $0.560 \pm 0.055$ |
| MONET$_G$ | **1.2** $\pm 1.7$ | $0.831 \pm 0.004$ |

Table 2: Results from the shilling attack experiment. Attackers attempt to insert 10 items in the top-20 recommendations of a target video. The results show that MONET can best mitigate the effect of an attack under incomplete information. We note that there is an implicit trade-off between debiasing and maintaining correlation with the original (biased) embeddings.

## 4.2 EXPERIMENT 2: THWARTING ATTACKS ON GRAPH-BASED RECOMMENDATION SYSTEMS

In this experiment we address H2, investigating the effectiveness of MONET to defend against a *shilling attack* (9) against graph-embedding based recommender systems (33). In a shilling attack, a number of users act together to artificially increase the likelihood that a particular influenced item will be recommended for a particular target item.

**Data.** In a single repetition of this experiment, we inject an artificial shilling attack into the MovieLens 100k dataset[3]. The raw data is represented as a bipartite graph with 943 users, 1682 items, and a total of 100,000 ratings (edges). Each user has rated at least 20 items. At random, we sample 10 items into an influence set $S_I$, and a target item $i_t$ to be attacked. We take a random sample of 5% of the existing users to be the set of attackers, $\mathcal{S}_A$. We then create a new graph, $G_{\text{attacked}}$ which in addition to all the existing ratings, contains new ratings from each attacker $\in \mathcal{S}_A$ to each item $\in \mathcal{S}_I$ as well as the target video. (Note that this corresponds to several varieties of behavior including both incentivizing formerly good users, and account takeover.)

**Design and Methods.** For each embedding method, we perform random walks through the new bipartite graph $G_{\text{attacked}}$. As we wish to study item recommendation, in the random walks, we simply remove user nodes each time they are visited (so the walks contain only pairwise co-occurrence information over items). With any given network embedding, we measure its bias by the number of influence items in $S_I$ in top-20 embedding-nearest-neighbor list of $i_t$. As metadata, we allow MONET models to know the per-movie attacker rating count for each attacked movie. However, to better demonstrate real-world performance, we only allow 50% (randomly sampled) attackers from the original 5% sample to be "known" when constructing these metadata. As non-debiasing baselines, we compare against DeepWalk and Glove. As debiasing baselines, we applied a generalized correlation removal framework developed for removing word embedding bias (27; 4). Specifically, we tried two approaches to "debias" the GloVe embedding of the MovieLens graph – as the "gender" embedding direction, we tried both (a) the most attacked movie vector and (b) the sum of attacked movie vectors. All methods use 128 dimensional topology embeddings and are trained on 100 random walks per node, each walk of length 5.

**Results.** As seen in Table 2, the topology embeddings from $\text{MONET}_G$ are the least biased by a large margin, letting on average only 1.2 influence items in the top-20 neighbors of $t_i$. Interestingly, we note that this behavior occurs even though the majority of observed co-occurrences for the algorithm had nothing to do with the attack in question, and only the known 50% of attackers were used to construct the metadata. $\text{MONET}_D$ had comparably less efficacy in this experiment. We speculate that this is due to a harmful effect of negative sampling on the learning of the metadata direction from the continuous attacker metadata - which would affect MONET's ability to debias the DeepWalk embeddings. Further research could investigate appropriate hyperparameter settings for $\text{MONET}_D$ in this case.

All other baselines (including those that explicitly model the attacker metadata) left at least around half of the attacked items in the top-20 list. To measure the extent to which debiased embeddings retain the original recommendation signal, we compute the pair-wise embedding distances of each method, and compute their Pearson correlation to the standard GloVe embeddings. We find that MONET embedding distances achieve high correlation (0.83) to the original distances, showing that with MONET it is possible to nearly nullify a shilling attack while preserving most of the signal from the true, un-attacked ratings. We note that the max-attack-embedding baseline higher embedding distance correlation, but this method let many more attacked items (on average) into higher ranks. This reveals a trade-off between embedding debiasing and prediction efficacy which has also been observed in other contexts (33).

## 5 RELATED WORK

Though graph learning is an immense field, a minority of unsupervised graph embedding techniques involve graph metadata. To our knowledge, none of these techniques involve either metadata orthogonalization or the capacity to learn arbitrary metadata transformations. (36) is a matrix factorization approach which uses a shared node embedding matrix to factor both the graph adjacencies and the raw metadata in a joint loss, with a tunable parameter to control the influence of the metadata loss. Similarly, (18) pre-computes a metadata similarity matrix and trains shared center-context embedding matrices on the metadata similarities and random walk similarities. In contrast, we learn the direction and effect of metadata as neural network parameters, and we separate those parameters into unique embedding dimensions. (31) and (35) are matrix factorization approaches which factor an approximation to the co-occurrence matrix into *equally-sized* metadata and topology embeddings, and were built mainly for text metadata. Their approaches enforce metric space similarity and

---

[3]Available: `http://files.grouplens.org/datasets/movielens/ml-100k/`

dimensional homogeneity between metadata and topology representations, restrictions that we do not rely on and are ill-suited to the setting with multiple types of arbitrarily-sized metadata. (16) constructs random walks that traverse between the original graph and the metadata freely, an approach which runs counter to our ability to separate out the effects of metadata on graph adjacencies. (20) introduce a version of the stochastic block model with metadata priors, and show that the estimated posteriors yield insight into the influence of metadata on the graph. However, this model estimates a community partition and in/out-community probabilities - it does not yield embeddings either of the node topology or the node metadata. There has been work in Natural Language Processing on removing gender bias from word embeddings (e.g. 4), but these methods operate with pre-computed embeddings and rely on identification of gendered terminology.

Additionally, there has been a wealth of work studying semi-supervised learning with graphs (e.g. 32) and graph convolutional networks (e.g. 2; 17; 11), which use graph metadata as features. While most semi-supervised and supervised neural networks for graphs indirectly produce embeddings that in some cases can be identified with feature and topology dimensions, they are trained as part of prediction or label propagation tasks. Therefore, the topology embeddings are free to correlate with features to the extent that this serves the loss function - there is no explicit separation of topology and metadata dimensions. In this paper, we have studied the benefits of metadata orthogonalization in the unsupervised setting, and we leave the exploration of our techniques in the semi-supervised and supervised settings to future work.

As described at the end of Section 3, there are many approaches to data representation learning that use adversarial networks to produce attribute-invariant embeddings. For instance, (30) and (13) use an adversary to allow feature embeddings to forget differences in data sources. Similar techniques have been applied to debias word embeddings from gender information (34) and recommender graph embeddings from demographic information (5). MONET differs from these approaches in a few key and consequential ways. First, (to our knowledge) none apply out-of-the-box to the unsupervised setting, which motivated our introduction of a baseline adversarial version of GloVe. Second, whereas MONET can accept any metadata in an appropriate design matrix $M$, different types of metadata require different adversary losses, which can require cumbersome tuning. Third, adversarial approaches induce a trade-off between accuracy on the main task and debiasing. In contrast, our work aims to minimize training error *subject to* perfect (linear) debiasing.

## 6 CONCLUSION

In this work, we have shown that unsupervised training of graph embeddings induces bias from important graph metadata. We proposed a novel solution to address this problem – the Metadata-Orthogonal Node Embedding Training (MONET) unit. The MONET unit is the first graph learning technique for training-time debiasing of embeddings, using orthogonalization. Our experimental results using real datasets showed that MONET is able to encode the effect of graph metadata in isolated embedding dimensions, and simultaneously remove the effect from other dimensions. This has immediate practical applications, which we illustrate by mitigating a simulated shilling attack on a real dataset of movie ratings.

We note that, because MONET only performs linear debiasing, the method is simply a first step in this area, and does not completely solve the problem of exact metadata independence. That being said, we argue that this is not a major limitation for many practical uses. As we show in the shilling attack experiment, MONET seems to greatly reduce bias when embeddings are used for simple nearest-neighbor lookups, which is a common application in graph-based recommender systems. Furthermore, advanced non-linear classifiers are not always scalable to graphs commonly found in industrial applications.

This work was meant to introduce the basic principles underlying the need for the MONET technique, and show its utility in a shallow graph neural network (GloVe). While we used a shallow network for instructional purposes, we note that MONET is generalizable, and MONET units can be used to debias any set of embeddings from another set during training. Subsequent research can explore the use of MONET in deeper networks and potentially semi-supervised models or graph convolutional networks. As MONET's SVD calculation can be expensive with large graphs and large embedding dimensions, future research could be in assessing the effect of SVD approximations, or training algorithms that utilize caching of previous metadata embedding SVDs to speed up training.

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

## A Appendix

**Proposition 1.** *Under the assumptions of Theorem 1, we have*

$$\mathbb{E}[Z^T\delta_W(i,j)] = 2\left[\Sigma_M(\Sigma_B - \Sigma_T) + \sigma_W I_m\right]\beta. \tag{6}$$

*Proof.* Derivatives of GloVe$^*_{meta}$ yield that the $i$-th row of $\delta_W(i,j)$ is $d_{ij}W_j^T$, where

$$\begin{aligned} d_{ij} &= \log(C_{ij}) - Z_i^T Z_j - W_i^T W_j - a_i - a_j \\ &= \theta + \tilde{Z}_i^T \tilde{Z}_j - Z_i^T Z_j - W_i^T W_j - a_i - a_j. \end{aligned} \tag{7}$$

Similarly the $j$-th row is $d_{ij}W_i^T$, and all other rows are zero vectors. Hence

$$\mathbb{E}[Z^T\delta_W(i,j)] = \mathbb{E}[Z_i d_{ij} W_j^T] + \mathbb{E}[Z_j d_{ij} W_i^T]. \tag{8}$$

We derive the second term on the right-hand side of Equation 8; the first term follows by symmetry. Note first that $\mathbb{E}Z_i(\theta - a_i - b_j)W_j^T = 0$ by independence and centering assumptions. Second:

$$\mathbb{E}[Z_i W_i^T W_j W_j^T] = T^T\mathbb{E}[M_i W_i^T W_j W_j^T] = T^T\mathbb{E}[M_i W_i^T]\mathbb{E}[W_j W_j^T] = T^T\beta\sigma_W I_d = T^T\sigma_W I_m\beta$$

by independence. Third:

$$\mathbb{E}[Z_i Z_i^T Z_j W_j^T] = T^T\mathbb{E}[M_i M_i^T TT^T M_j W_j^T] = T^T\left(\mathbb{E}[M_i M_i^T]\right)TT^T\left(\mathbb{E}[M_j W_j^T]\right) = T^T\Sigma_M\Sigma_T\beta$$

by independence, and similarly $\mathbb{E}[Z_i \tilde{Z}_i^T \tilde{Z}_j W_j^T] = T^T\Sigma_M\Sigma_B\beta$. Combining these with Equation 7, we have

$$\mathbb{E}[Z_i d_{ij} W_j^T] = T^T\left[\Sigma_M(\Sigma_B - \Sigma_T) - \sigma_W I_d\right]\beta. \tag{9}$$

Applying symmetry to the second term in Equation 8 completes the proof. □

### A.1 Proof of Theorem 1

*Proof.* Proposition 1 gives $\mathbb{E}[Z^T\delta_W(i,j)] = 2T^T\left[\Sigma_M(\Sigma_B - \Sigma_T) + \sigma_W I_m\right]\beta$. Second, note that $\mathbb{E}[M_i W_i^T] = \beta \Rightarrow M^T W = n\beta$ and thus $Z^W = T^T M^T W = nT^T\beta$. Recalling that $W' = W + \delta_W(i,j)$ we have

$$\mathbb{E}[Z^T W'] = 2\left[\Sigma_M(\Sigma_B - \Sigma_T) + (n - \sigma_W)I_m\right]\beta.$$

Applying Jensen's Inequality completes the proof. □

### A.2 Proof of Theorem 2

*Proof.* Consider metadata embeddings $Z \in \mathbb{R}^{n \times d_Z}$ and, as in the MONET algorithm, define the projection $P_Z = I_{d_Z} - Q_Z Q_Z^T$, where $Q_Z$ are the left-singular vectors of $Z$. By properties of the SVD, $Z^T Q_Z Q_Z^T = Z_T$, and hence $Z^T P_Z = \mathbf{0}_{n \times d_Z}$. This means that $Z^T W^\perp = Z^T\delta_W^\perp = \mathbf{0}_{d_Z \times d}$, which completes the proof by definition of metadata leakage. □

| Model | Wall Time (sec) | SVM Accuracy | Embedding Distance Correlation w/GloVe |
|---|---|---|---|
| Random | N/A | $0.527 \pm 0.000$ | $0.004 \pm 0.001$ |
| DeepWalk | $48.562 \pm 0.579$ | $0.896 \pm 0.000$ | $0.630 \pm 0.006$ |
| GloVe | $98.487 \pm 7.220$ | $0.948 \pm 0.000$ | $1.000 \pm 0.000$ |
| GloVe$_{adversary}$ | $162.575 \pm 7.042$ | $0.549 \pm 0.018$ | $0.432 \pm 0.02$ |
| GloVe$_{meta}$ | $102.387 \pm 7.595$ | $0.906 \pm 0.000$ | $0.862 \pm 0.020$ |
| MONET$_D$ | $595.477 \pm 29.66$ | $0.796 \pm 0.032$ | $0.035 \pm 0.005$ |
| MONET$_G$ | $112.505 \pm 7.156$ | $0.899 \pm 0.000$ | $0.334 \pm 0.017$ |

Table 3: Additional metrics from the political blogs experiment. Wall time is the user wait time during training. SVM Accuracy is the accuracy on a non-linear SVM with an RBF kernel. Embedding Distance Correlation with GloVe is the Pearson correlation of the pairwise embedding distances between each set of embeddings and GloVe embeddings.

## A.3 DESCRIPTION OF MONET$_D$: MONET IMPLEMENTED IN DEEPWALK

The implementation of MONET$_D$ follows the general procedure laid out in Algorithm 1. As with MONET$_G$, for DeepWalk the MONET unit adds a feed-forward transformation of the metadata to the graph representation, resulting in metadata embeddings $X$ and $Y$ (see Eq. 3). $Z = X + Y$ gives the combined metadata representation, used to debias $U$ and $V$ via $P_Z$ (see Algorithm 1 and surrounding description). MONET does not affect the negative sampling component of DeepWalk's loss (Eq. 3). MONET debiases the topology embeddings as described, which are then used throughout the standard DeepWalk model (along with the metadata embeddings).

## A.4 ADVERSARIAL BASELINE

As our adversarial baseline, we implemented a 2-layer MLP discriminator following the framework of (14). The MLP had ReLU activations and an 8 dimensional hidden layer. The MLP was trained to predict political party from the topology vectors of each batch of input nodes, using cross-entropy loss (discriminator task). Then, the topology vectors were fed through the discriminators, but their negative logits were used for prediction (topology task). The discrimnator task and topology task were evaluated and optimized after each optimization of the GloVe loss.

## A.5 ADDITIONAL RESULTS FROM POLITICAL BLOGS EXPERIMENT

In Table 3 we give the following additional metrics computed from the political blogs experiment, averaged over thirty repetitions:

1. **Wall Time (sec)**: user wait time in seconds from the beginning to end of each method. The significant increase seen from MONET$_D$ is due to DeepWalk's negative sampling loss.

2. **SVM Accuracy**: to emphasize the fact that MONET only performs *linear* debiasing, we trained a non-linear SVM with an RBF kernel using a randomly-chosen 50% of the nodes as training points.

3. **Embedding Distance Correlation w/GloVe**: This metric was also used in the shilling experiment. In our experiments, GloVe$_{meta}$ and MONET$_G$ were implemented into the GloVe model, so we use the correlation between the pairwise embedding distances between MONET models and GloVe to measure the amount that metadata embedding and SVD correction has "corrupted" the embeddings. As the results show, among non-random methods, MONET$_G$ and MONET$_D$ are most corrupted. This is because most of the community signal in the political blogs network is due to the political affiliation attribute, and MONET models explicitly removes linear correlation with that attribute. That said, we see that MONET models still preserve a statistically significant amount of signal from the GloVe embeddings above the random baseline.

Additionally, we perform a variant of the political blogs experiment to test the inductive performance of MONET. Specifically, we aim to answer the question: if some nodes are not used to train the MONET model, can we still use the MONET model to debias other embeddings for those nodes? Given a subset of nodes $N_a$, we train a MONET model on the induced subgraph $G_a := G(N_a)$. We then attempt to debias the DeepWalk embeddings of held-out nodes $N_b := N \setminus N_a$. To do this, we apply the MONET metadata transformation $T_1, T_2$, trained on $G_a$, to the metadata of $N_b$. This yields

inductive metadata embeddings $Z_b$ for the nodes $N_b$. We then apply the de-biasing projection given in Algorithm 1 to the DeepWalk embeddings for $N_b$, and compute the linear SVM Macro/Micro-F1 scores to measure embddings bias.

Figure 4 shows the average Macro/Micro-F1 over 5 repetitions per proportion of held-out nodes. For reference, we also give the accuracy scores for the un-corrected DeepWalk embeddings for $N_b$. Note that the linear SVM was trained on a 50% training-test split across $N_b$ only. We find that the MONET transformation learned from the given nodes is able to properly generalize to and debias the new data $N_b$.

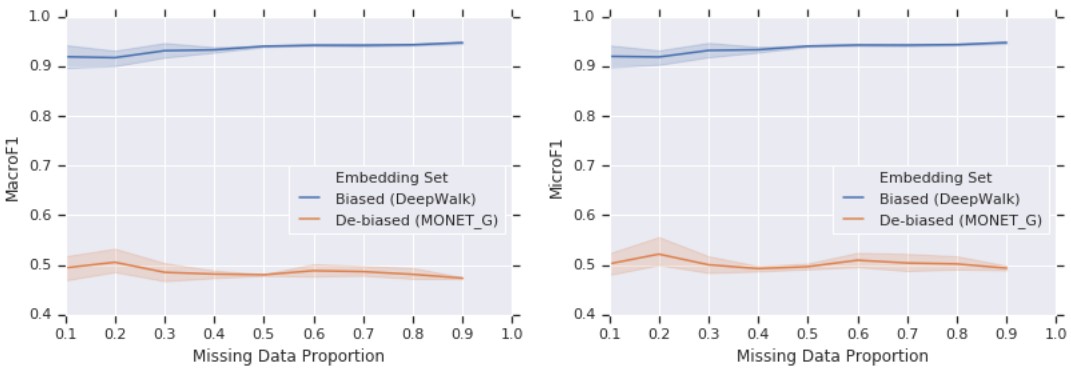

Figure 4: Macro/Micro-F1 scores from a linear SVM classifier, trained on "held-out" node embeddings from both DeepWalk and inductive MONET.

