# OpenReview forum: "MONET: Debiasing Graph Embeddings via the Metadata-Orthogonal Training Unit"
_ICLR.cc/2020/Conference — Reject_

### Official Review · AnonReviewer1 · 2019-10-23
**Official Blind Review #1**

**Rating:** 6

**Review:**

Summary: The paper introduces a GNN model (MONET) for debiasing graph embeddings, by enforcing orthogonality between the embedding spaces of the graph topology & the graph metadata. They show that unsupervised learning induces bias from important graph metadata, when the metadata is correlated with the node edges. They show experimental results on real world graphs (political blogs network & graph-based recommendation systems), where MONET can debias  graph embeddings and prevent metadata leakage.

Decision: Accept

Reasons for the decision: The paper is clearly written, well-motivated, and well-organized. The proposed algorithm and analysis seem insightful & novel, and the experimental results (showing that MONET can debias metadata from topology) are convincing.

Additional Feedback:

1) It would be helpful to show results on at least one other graph embedding model other than GloVe, to empirically substantiate the claim that MONET is “broadly generalizable”.

2) In Section 3.4 [Algorithmic Complexity], it would be helpful to compare the wall clock time of MONET vs. the baselines (DeepWalk, GloVe), to give a better sense of how expensive the SVD calculation is.


**Experience Assessment:**

I do not know much about this area.

**Review Assessment: Checking Correctness Of Derivations And Theory:**

I did not assess the derivations or theory.

**Review Assessment: Checking Correctness Of Experiments:**

I assessed the sensibility of the experiments.

**Review Assessment: Thoroughness In Paper Reading:**

I read the paper at least twice and used my best judgement in assessing the paper.

---

> ### Author Response · Authors · 2019-11-13
> **Author Response 1**
>
> We appreciate the suggestions. In the Appendix, we now display the wall clock time for all methods involved in the political blogs experiment. We added a reference to that table in Section 3.4.
>
> Regarding results from another graph embedding model with MONET, we are currently working on an implementation of DeepWalk with MONET, which we hope to have completed by the end of the discussion period. Note that we have described the MONET version of DeepWalk in Section 3.1 (Equation 3).

---

> ### Author Response · Authors · 2019-11-15
> **Author Response 2: addition of MONET-DeepWalk implementation**
>
> Fortunately, we were able to complete the implementation of MONET into the DeepWalk loss. Results from the new method MONET_D are now compared in the experiments, and the implementation is explained fully in  the Appendix.
> The code we will release includes implementations of both MONET_G and MONET_D.

---

### Official Review · AnonReviewer2 · 2019-10-23
**Official Blind Review #2**

**Rating:** 6

**Review:**

The paper presents an approach to debiasing graph embeddings from known, given node attributes/metadata.
Specifically, the paper proposes to learn an embedding that is orthogonal to given node attributes, ensuring that there is no *linear* function which can extract the node attributes from the learned embedding.


Strength:
-	The paper addresses an interesting and relevant problem on debiasing graph embeddings.
-	The paper presents a, to my knowledge, novel approach, to avoid the leakage of meta-data in the embedding, effectively debiasing it from this information (although some discussion of prior should be addressed, see below)
-	The paper shows that the approach is effective compared to two baselines on two datasets (although some aspects of the experiments can be improved, see below)

Weaknesses:
1.	Experimental evaluation:
1.1.	The paper only evaluates on the training set. Specifically, in experiment 1, the paper learns an embedding and supervises it to be orthogonal to given labels; this is good, but it also somewhat expected that this could be learned when tested on the same dataset. An important question is, if the model actually learned a generalizable embedding or just overfit to the training set. If the learned embedding is applied to new data, is it still not possible to extract political affiliation from it?
I would suggest splitting the dataset in two part: One part to train the debiased embedding and one to train and test the Linear SVM.
1.2.	It would be great if the authors describe better and quantify how they ensure in experiment 1 that the learned embedding of the MONET model is measured, and how this compares to the three baselines (e.g. a random or constant embedding would also be perfectly debiased).
1.3.	While the paper clearly states that the approach is restricted to linear relationships, it would be interesting to look at non-linear classifiers and see how well this works in practice, also in comparison to the baselines.
1.4.	Figure 3(c) visualizes that
2.	Related work:
2.1.	The comparison to related work could be improved. Specifically, a discussion relating this work to adversarial training, e.g. as in domain confusion networks [A] or in [5].
2.2.	The authors argue that [5] is independent/concurrent work. I agree with the authors that [5] is sufficiently different to this work, but it should be discussed thoroughly as it has been published at ICML 2019. Unfortunately, the authors also miss to include in the references that it has been published at ICML 2019.


While the paper explores an interesting direction and approach, there are several concerns which speak against acceptance (see Weaknesses above); however, I believe they can be a addressed/clarified in a further revision.


References:
[A] Tzeng et al, Adversarial Discriminative Domain Adaptation, CVPR 2017


=== Post author response
I thank the authors for their response and additions as well as clarifications.

I agree to other reviewers, that the limitation to linear de-biasing is a concern for the paper, but the authors have clarified it now in the abstract an other locations; the additional experiments with and RBF kernel have shown that  indeed the formulation only does mainly do linear decorrelation.

Table 3 could be further clarified (what is the goal? 50% accuracy, i.e. chance level?) and made more similar to Table 1, or ideally merged with it.

It is also a bit unfortunate that all this additions are in supplement and not merged in the main paper.

Overall I still lean more towards accept.

**Experience Assessment:**

I do not know much about this area.

**Review Assessment: Checking Correctness Of Derivations And Theory:**

I assessed the sensibility of the derivations and theory.

**Review Assessment: Checking Correctness Of Experiments:**

I carefully checked the experiments.

**Review Assessment: Thoroughness In Paper Reading:**

I read the paper at least twice and used my best judgement in assessing the paper.

---

> ### Author Response · Authors · 2019-11-13
> **Author Response 1**
>
> We appreciate the comments and interesting considerations. To examine whether MONET can learn a "generalizable embedding", we split the graph into two parts, and train a MONET model on the first part. We then use the metadata transformation from that restricted MONET model to debias the usual DeepWalk embeddings of those nodes. In the Appendix, we show that this process is able to debias the held-out data effectively, evaluating at a full range of sizes for the held-out set.
>
> The issue of embedding quality after debiasing is a good question. We have added the "Embedding Distance Correlation to GloVe" metric to the political blogs experiment (now shown in the Appendix). This measures the correlation of pairwise embedding distances between GloVe and the various other methods. We have also added a random embeddings baseline to that experiment. We find that MONET embedding distances are significantly more correlated to GloVe distances than the random baseline, and that the random baseline is actually significantly biased, on average (randomly generated embeddings are never perfectly linearly separated from any given sensitive attribute).
>
> On the political blogs data, we have also added political affiliation classification results using a non-linear SVM (using an RBF kernel), in response to your question about the application of non-linear classifiers. We find that, while MONET is still less biased than the baselines, the gap diminishes drastically, and the classifier performs quite well even on MONET embeddings.
> We think that researching a non-linear correction in the style of MONET is an exciting area for future work.
>
> We have now properly cited Bose and Hamilton (2019), and added a substantial discussion of related work from the field of adversarial learning.

---

### Official Review · AnonReviewer3 · 2019-10-29
**Official Blind Review #3**

**Rating:** 3

**Review:**

TLDR: split node embeddings into medatadata and graph structure, force them to be orthogonal.

The paper proposes to split node embeddings in a graph into two parts:
1. graph structure embeddings: Es
2. known node metadata embeddings: Em
To prevent Es from containing information about Em, the authors propose a scheme which puts Es into the Nullspace of Em through repeated SVD factorizations. This prevents linear classifiers that operate on Es to reliably predict information in Em.

The weakness of the paper stems from the proposed definition of debiasing. Just like two random variables can be dependent, but have a linear correlation coefficient of 0, in the proposed method the two embeddings may be linearly unrelated, but have a strong non-linear relationship.

This is an important caveat that should be highlighted in the papers' abstract, not burried deep on p4, under Theorem 2.

In fact, looking at Fig 3c information about party affiliation follows a XOR-like pattern in the PCA space. This means that a linear classifier will fail (indeed the linear SVM in Table 1 fails), but a non-linear one should work OK. Thus, contrary to the abstract, the proposed method doesn't remove the effect of arbitrary covariates, but removes a LINEAR dependence.

Thus the paper proposes to solve an important problem and proposes a partial solution, but overstates the results in the abstract and hides the true efficiency of the method.

Action items ot correct the paper:
- be more honest about the true result. Decorrelation does not imply independence.
- redo Table 1 with strong non-linear classifiers such a Gaussian SVM or Random Forest to show how much is not filtered out by your linear decorrelation method

Finally, contrast with the adversarial information removal [1] and  the information bottleneck [2], both of which also promise to remove non-linear dependencies. It may happen that the you method works better, even though it only guarantees no linear dependencies.

[1] https://arxiv.org/abs/1505.07818
[2] D. Moyer, S. Gao, R. Brekelmans, A. Galstyan, and G. Ver Steeg, “Invariant Representations without Adversarial Training,” in Advances in Neural Information Processing Systems 31, 2018



**Experience Assessment:**

I have read many papers in this area.

**Review Assessment: Checking Correctness Of Derivations And Theory:**

I assessed the sensibility of the derivations and theory.

**Review Assessment: Checking Correctness Of Experiments:**

I assessed the sensibility of the experiments.

**Review Assessment: Thoroughness In Paper Reading:**

I read the paper at least twice and used my best judgement in assessing the paper.

---

> ### Author Response · Authors · 2019-11-13
> **Author Response 1**
>
> We appreciate the careful criticisms and the deep understanding of our work. We have made sure to clearly state the linear nature of MONET debiasing in the abstract, in our list of contributions in the introduction, and in our conclusion. We have also changed statements about the "independence" of metadata and topology embeddings to "decorrelated". We agree, as you wrote, that MONET is a partial solution to the problem of complete debiasing of unsupervised graph embeddings. That being said, we argue that MONET remains a useful first step in this area. As we show in the shilling attack experiment, MONET substantially reduces bias when embeddings are used for simple nearest-neighbor lookups, which is a common application in graph-based recommender systems. We have added a discussion about this to our conclusion.
>
> Additionally, in response to your suggestions and those of the second reviewer, we have added results from a non-linear SVM applied to political blog embeddings, showing that indeed non-linear classifiers still work well on MONET-debiased embeddings. We added a reference to these metrics in the main text.
> We look forward to investigating non-linear corrections in the style of MONET in our future work in the area.
>
> We like your suggestion to compare against adversarial or information bottleneck baselines.  We looked into the code bases for the references you listed (and others), but couldn't find any off-the-shelf implementations for unsupervised debiasing that were directly applicable to our problem.  Are you aware of any?
>
> In lieu of a reusable baseline, we have started adapting a couple existing methods that seemed close ([1], [2]).  Unfortunately this has gone slowly - the adversarial methods have been non-trivial to adapt to our setting, and cumbersome to train.  We can add the current results to the manuscript, but we would prefer to wait and tune the baselines further. (We will complete this ASAP, but likely not before the rebuttal period is over.)
>
>
> [1] A. J. Bose and W. L. Hamilton.  Compositional fairness constraints for graph embeddings. Proceedings of the 36th International Conference on Machine Learning, 2019.
>
> [2] B. H. Zhang, B. Lemoine, and M. Mitchell.   Mitigating unwanted biases with adversarial learning. In Proceedings of the 2018 AAAI/ACM Conference on AI, Ethics, and Society, pages335–340. ACM, 2018.

---

> ### Author Response · Authors · 2019-11-15
> **Author Response 2: addition of adversarial baseline**
>
> Fortunately, we were able to implement an attribute adversary into our GloVe model using the framework in [1]. All references to adversarial learning work you gave (and others mentioned by other reviewers) are extremely relevant, and aim to solve closely related problems. We discuss them in our related work section. However, none of the associated models apply directly to our problem, unless one makes significant modifications (verging on novel work). Therefore we decided to directly adapt a fundamental adversarial learner (from [1]) to our GloVe trainer. To our knowledge, this is actually a novel (though naive and preliminary) incorporation of adversarial learning in unsupervised GNNs. While by no means a mature and complete approach to the problem, we believe it helps shed light on how a comparable adversarial approach would work as an alternative to MONET. The option to use the adversarial loss will be available in the code we are open sourcing as a reference implementation of the ideas discussed in the paper.
>
> Since our last update we have added a paragraph about the connection to adversarial networks in our Methods section (S 3.4). We now describe our adversarial baseline in the beginning of the experiments section and more fully in the Appendix. Note that the shilling experiment involves real-valued attributes, which requires a different type of adversarial network than that needed for the political blogs experiment. We consider the development of multiple adversaries in our unsupervised setting beyond the scope of this paper, so we use the adversary baseline only for the first experiment.
>
>
> [1] I. Goodfellow, J. Pouget-Abadie, M. Mirza, B. Xu, D. Warde-Farley, S. Ozair, A. Courville, and Y. Bengio. Generative adversarial nets. In Advances in neural information processing systems, pages 2672–2680, 2014.

---

### Decision · Program_Chairs · 2019-12-19

**Decision:**

Reject

**Comment:**

This work presents a method for debiasing graph embeddings. The main concerns for the work were originally identified by Reviewer 3, who pointed out that the method is only capable of linear debiasing. Authors responded by updating the manuscript in several places to mention this limitation as well as adding Table 3 to the Appendix showing that SVM's with non-linear kernels are still able to identify bias in the embeddings. Reviewers agreed that this addition improved the manuscript, however some reviewers still had concerns about the revised manuscript. This AC has several recommendations for improving the paper. First additional revision is needed to better address the limitations of linear debiasing, for example Table 1 still reads "MONET is successful in removing all metadata information from the topology embeddings – the links in the graph are no longer an effective predictor of political party".  Statements like this are a bit misleading, as the embeddings will still be biased with respect to a non-linear classifiers (as evident by Table 3). Additionally, updating Table 1 and related experiments to measure embedding bias with respect to non-linear classifiers would help clarify the limitations for perspective readers. Second, the paper should be updated to address remaining concerns that the linear debiasing assumption limits the applicability of the method. One could either discuss or demonstrate additional applications of the method that work even with the linear assumption, extend MONET so it can improve model bias with respect to non-linear classifiers, or show that MONET still outperforms baselines when the non-linear assumption is violated.